# Isolation and Identification of Pentalenolactone Analogs from *Streptomyces* sp. NRRL S-4

**DOI:** 10.3390/molecules26237377

**Published:** 2021-12-05

**Authors:** Huanhuan Li, Hongji Li, Shuo Chen, Wenhui Wu, Peng Sun

**Affiliations:** 1Department of Phytochemistry, School of Pharmacy, Second Military Medical University, 325 Guo-He Road, Shanghai 200433, China; m13083766175@163.com (H.L.); lihongji0327@smmu.edu.cn (H.L.); m190310884@st.shou.edu.cn (S.C.); 2Department of Marine Bio-Pharmacology, College of Food Science and Technology, Shanghai Ocean University, 999 Huchenghuan Road, Shanghai 201306, China

**Keywords:** *Streptomyces*, pentalenolactone, terpenoid, genome mining, secondary metabolites

## Abstract

Terpene synthases are widely distributed in Actinobacteria. Genome sequencing of *Streptomyces* sp. NRRL S-4 uncovered a biosynthetic gene cluster (BGC) that putatively synthesizes pentalenolactone type terpenes. Guided by genomic information, the S-4 strain was chemically investigated, resulting in the isolation of two new sesquiterpenoids, 1-deoxy-8*α*-hydroxypentalenic acid (**1**) and 1-deoxy-9*β*-hydroxy-11-oxopentalenic acid (**2**), as shunt metabolites of the pentalenolactone (**3**) biosynthesis pathway. Their structures and absolute configurations were elucidated by analyses of HRESIMS and NMR spectroscopic data as well as time-dependent density functional theory/electronic circular dichroism (TDDFT/ECD) calculations. Compounds **1** and **2** exhibited moderate antimicrobial activities against Gram-positive and Gram-negative bacteria. These results confirmed that the pentalenolactone pathway was functional in this organism and will facilitate efforts for exploring Actinobacteria using further genome mining strategies.

## 1. Introduction

Actinobacteria have been proven to be one of the most reliable sources of natural products with industrial and medicinal importance. Whole-genome sequencing and bioinformatics analyses reveal an enormous number of secondary metabolite biosynthetic gene clusters (BGCs) in Actinobacteria [1]. Given that genes encoding terpene synthases have been found to be widely distributed [2], it is believed that the amount of terpene metabolites currently reported in Actiobacteria is an underestimation. Genome mining approaches have greatly enhanced the discovery of secondary metabolites from Actinobacteria [3]. In the course of our ongoing search for bioactive metabolites from marine invertebrates and microorganisms [4,5,6], the strain *Streptomyces* sp. NRRL S-4 (S-4) was selected for the exploration of secondary metabolite profiles. Previous studies of this strain resulted in the discovery of thiostreptamides and venturicidins [6,7]. Whole-genome sequencing revealed that S-4 contained 25 BGCs including six BGCs that putatively synthesized the terpenoids [6]. Aiming at terpenoid metabolites, we chemically investigated the organic extracts of S-4 fermentation and discovered two new sesquiterpenoids, 1-deoxy-8*α*-hydroxypentalenic acid (**1**) and 1-deoxy-9*β*-hydroxy-11-oxopentalenic acid (**2**). Herein, we reported the isolation, elucidation, and biological activities of these compounds.



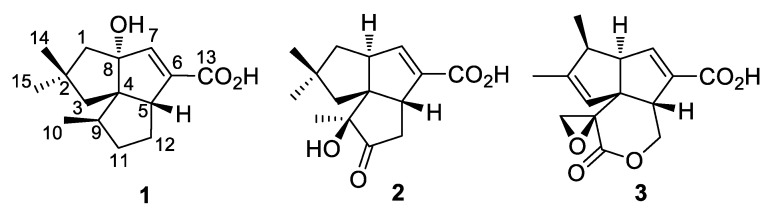



## 2. Results

### 2.1. Pentalenolactone BGC in S-4

The whole genome of S-4 was 7.4-Mb in size. With online antiSMASH analysis, a total of 25 putative secondary metabolite BGCs were predicted including six terpene BGCs (Appendix A) [8]. One terpene BGC, here designated as the *pll* cluster, possibly synthesized the pentalenolactone (**3**) metabolites. The 12.9-kb *pll* BGC consisted of 10 open reading frames (ORFs) showing an average >90% sequence similarity to those of the *pnt* BGC from validated pentalenolactone producer *S. arenae* [9]. (Table 1) The ORFs of the *pll* BGC were identical in organization to those of the *pnt* BGC. In silico genome analysis implied the possibility of finding pentalenolactone or its derivatives in the S-4 culture broth.

### 2.2. Isolation and Structural Identification

Compound **1** was isolated as an amorphous powder. Its molecular formula was determined to be C_15_H_22_O_3_ on the basis of high-resolution electrospray ionization mass spectrometry (HRESIMS) at *m/z*: 249.1489 [M − H]^−^ (calcd for C_15_H_21_O_3_, 249.1485), requiring five double-bond equivalents (DBE). The IR spectrum displayed absorptions for hydroxy (3377 cm^−1^) and *α*, *β*-unsaturated carboxylic acid (1684 cm^−1^) functionalities. The ^13^C-NMR and distortionless enhancement by polarization transfer (DEPT) spectra displayed 15 signals which corresponded to 3 sp^2^ (1 C=O, 1 C=CH) and 12 sp^3^ (3 CH_3_, 4 CH_2_, 2 CH, 1 OC, 2 C) carbon atoms, accounting for two degrees of unsaturation (Table 2 and Appendix A). The remaining DBEs are indicative of three rings in the molecule. The olefinic proton and two sp^2^ carbon signals (*δ*_H_ 6.41, H-7; *δ*_C_ 148.0, C-7; *δ*_C_ 140.1, C-6) were readily recognized as markers for a trisubstituted double bond. The HMBC cross-peak from H-7 to C-13 established an *α*, *β*-unsaturated carboxylic acid moiety (Figure 1). The HMBC correlations from H-7 to C-4, C-5, C-6, and C-8, and from allylic proton (*δ*_H_ 3.02, H-5) to C-6 permitted the assignment of ring B. Analysis of the COSY spectrum characterized a proton spin system of H_3_-10/H-9/H_2_-11/H_2_-12/H-5. The HMBC cross-peaks from H_3_-10 to C-4, C-9, and C-11, and from H-9 to C-5 built ring C. The HMBC correlations from two geminal methyls of H_3_-14 and H_3_-15 to C-1, C-2, and C-3, from H_2_-1 to C-4 and C-8, and from H_2_-3 to C-4 and C-5 established the third cyclopentane (ring A).

In the NOESY spectrum of **1**, the observed NOE correlations of H-1*β*/H_3_-14, H-1*β*/H-7, H_3_-14/H-3*β*, H-3*β*/H-5, H-5/H_3_-14, H-5/H-12*β*, and H-12*β*/H-11*β* indicated a *β* orientation of these protons (Figure 1). NOE cross-peaks of H-1*a*/H_3_-15, H-9/H-11*a*, and H-11*a*/H-12*a* revealed that these protons were *α* oriented. A (4*S**,5*R**,8*R**,9*R**)-**1** relative configuration was then deduced. The absolute configuration was determined by time-dependent density functional theory/electronic circular dichroism (TDDFT/ECD) calculation. The Boltzmann-weighted ECD spectrum showed a good superposition with the experimental curve of **1**, particularly, the positive band at 208 nm and the negative band at 247 nm (Figure 2 and Appendix A). Therefore, **1** was assigned to have a (4*S*,5*R*,8*R*,9*R*) configuration and named as 1-deoxy-8*α*-hydroxypentalenic acid. 

Compound **2**, a white powder, has a molecular formula of C_15_H_20_O_4_ as determined by HRESIMS at *m/z* 263.1282 [M − H]^−^ (calcd for C_15_H_19_O_4_, 263.1278). The ^1^H and ^13^C-NMR spectra of **2** showed a close similarity to those of **1**, suggesting a sesquiterpenoid skeleton (Table 2 and Appendix A). A difference was observed with the presence of a ketone group (*δ*_C_ 217.1, C-11) in the shielded region of the ^13^C spectrum and three methyl singlets in the deshielded region of the ^1^H spectrum. The COSY spectrum indicated the presence of two isolated proton sequences, of H_2_-1/H-8/H-7 and H-5/H_2_-12 (Figure 3). The HMBC correlations from H_3_-10 to C-4, C-9 (*δ*_C_ 80.5), and C-11, and from H_2_-12 to C-11 located the ketone group at C-11 and the oxygen-bearing quaternary carbon at C-9. The HMBC correlations from H-7 to C-4, C-5, and C-13 confirmed the presence of *α*, *β*-unsaturated carboxylic acid and a B ring. The HMBC cross-peaks from both H_3_-14 and H_3_-15 to C-1, C-2, C-3, and C-4, and from H_2_-1 to C-4 and C-7, and from H_2_-3 to C-4 and C-5 confirmed the presence of a complete ring A.

Compound **2** showed a similar NOE pattern to that of **1**. The NOE correlations of H-1*β*/H_3_-14, H-1*β*/H-7, H-3*β*/H_3_-14, H-3*β*/H-5, H-5/H_3_-14, H-5/H-12*β*, and H-7/H_3_-14 suggested that these protons were in *β* orientation. The NOE cross-peaks of H-1*a*/H_3_-15, H_3_-15/H-3*a*, H-3*a*/H_3_-10, H-1*a*/H-8, and H-8/H-10 indicated *α* orientation of these protons. The 10-OH group was then deduced to be in *β* orientation. The relative configuration of **2** was assigned to be (4*R**,5*R**,8*S**,9*R**). The structure of **2** was analyzed with the TDDFT/ECD calculation. The calculated ECD spectrum of **2** exhibited a high similarity with the experimental curve (Figure 4 and Appendix A), which led to the assignment of an absolute configuration of **2** of (4*R*,5*R*,8*S*,9*R*). Compound **2** was assigned to be 1-deoxy-9*β*-hydroxy-11-oxopentalenic acid.

### 2.3. Antimicrobial Activity

The two compounds were subsequently evaluated for possible antibacterial activity. Compounds **1** and **2** showed moderate antibacterial activities against Gram-positive bacteria *Staphylococcus aureus* ATCC 25923 with MICs of 16 and 16 μg/mL, and against Gram-negative bacteria *Escherichia coli* ATCC 25922 with MICs of 32 and 16 μg/mL. Kanamycin and ampicillin were used as positive controls with 4 and 2 μg/mL for *S. aureus* and 16 and 4 μg/mL for *E. coli*, respectively. The results indicated that the epoxylactone moiety of pentalenolactone played an important role in the antimicrobial activity.

### 2.4. Proposed Biosynthetic Pathway

Pentalenolactone (**3**) is a widely occurring sesquiterpenoid antibiotic that has been isolated from a variety of *Streptomyces* species [9,10,11,12]. It exhibits antimicrobial action against bacteria, fungi, and protozoa by a reaction of the electrophilic epoxylactone moiety with the active site cysteine of glyceraldehyde-3-phosphate dehydrogenase [13]. Several pentalenolactone BGCs, e.g., *pen* from *S. exfoliatus*, *pnt* from *S. arenae*, and *ptl* from *S. avermitilis* and biochemical functions of ORFs as well as the biosynthetic pathway have been well characterized by Cane and collaborators (Figure 1) [14,15,16,17]. A high number of biosynthetic intermediates and shunt metabolites in the conversion from farnesyl pyrophosphate (FPP) to pentalenolactone have been isolated from several pentalenolactone producers [14,18,19]. Compounds **1** and **2** can be determined to be in the main pentalenolactone biosynthetic pathway as shunt metabolites. Biosynthetically, PntH, a non-heme iron, *α*-ketoglutarate-dependent hydroxylase, is responsible for catalyzing 1-deoxypentalenic acid to 1-deoxy-11*β*-hydroxypentalenic acid [16]. Compound **1** was assumed to be a shunt metabolite via 8*α* hydroxylation of 1-deoxypentalenic acid. Compound **2** might be derived from 1-deoxy-11-oxopentalenic acid via endogenous biotransformation during the fermentation process. A similar phenomenon occurs with pentalenic acid, a common pentalenolactone shunt metabolite isolated from many *Streptomyces* species. In *S. avermitilis*, a cytochrome P450 (CYP105D7) encoded by *sav7469*, which is not present in the *ptl* cluster, was found to be responsible for the conversion of 1-deoxypentalenic acid to pentalenic acid [20]. It is likely that there are some CYPs in S-4 responsible for the formation of **1** and **2**. As the final product, pentalenolactone, was not isolated from S-4, the function of PllE, a predicted Baeyer–Villiger monooxygenase (BVMO), was bioinformatically examined. The amino acid sequence of PllE was shown to have 91% identity and 96% similarity to the orthologous PntE [9]. The fingerprint motif FxGxxxHxxxWP/D for type I BVMO and the sequence motif GxGxxG for FAD and NADPH cofactors binding was preserved in PllE [21]. The reason the final pentalenolactone product was not detected might be partially explained by the instability of the epoxylactone moiety in pentalenolactone, or by the methods used to process fractions of S-4. The identity of the enzymes responsible for the formation of **1** and **2** requires further investigation.

## 3. Materials and Methods 

### 3.1. Fermentation and Isolation

The strain S-4 was inoculated in a 250 mL Erlenmeyer flask containing 100 mL of ISP2 medium and then grown on 10 L of fermentation medium (soybean flour 10 g/L, glucose 10 g/L, soluble starch 15 g/L, yeast extract 5 g/L, NaCl 5 g/L, CaCO_3_ 3 g/L, pH 7.0) at 28 °C for 7 days. Using the same isolation procedure [6], the EtOAc extract of S-4 was subjected to CC on ODS to afford 13 fractions (Frs. 1–13). Fr.8 was fractionated by Sephadex LH-20 (CH_2_Cl_2_/MeOH, 2:1) followed by RP-HPLC (60% MeOH, 1.5 mL/min) to give compounds **1** (6.0 mg, *t*_R_ 40.3 min) and **2** (17.2 mg, *t*_R_ 36.1 min).

Compound **1**: white amorphous powder; *R*_f_ 0.39 (CH_2_Cl_2_/MeOH 15:1); [α]^28.7^_D_ -30 (*c* 0.1, MeOH); UV (MeCN) *λ*_max_ (log *ε*) 213 (3.45) nm; ECD (*c* 3.1 × 10^−5^, MeCN) *λ*_max_ (Δ*ε*) 208 (15.9), 247 (-6.3) nm; IR (film) *ν*
_max_ 3377, 2951, 2926, 1689, 1366, 1251, 1053, 755 cm^−1^. ^1^H and ^13^C-NMR data see Table 2; HRESIMS *m/z*: 249.1489 [M − H]^−^ (calcd for C_15_H_21_O_3_, 249.1485), 273.1512 [M + Na]^+^, 233.1587 [M + H -H_2_O]^+^.

Compound **2**: white amorphous powder; *R*_f_ 0.40 (CH_2_Cl_2_/MeOH 15:1); [α]^29.3^_D_ +8.4 (*c* 0.05, MeOH); UV (MeCN) *λ*_max_ (log *ε*) 220 (3.69) nm; ECD (*c* 3.1 × 10^−5^, MeCN) *λ*_max_ (Δ*ε*) 203 (−18.6), 227 (24.9) nm; IR (film) *ν*
_max_ 3385, 2952, 2932, 1684, 1367, 1216, 1080, 750 cm^−1^. ^1^H and ^13^C-NMR data see Table 2; HRESIMS *m/z*: 263.1282 [M − H]^−^ (calcd for C_15_H_19_O_4_, 263.1278), 287.1325 [M + Na]^+^, 247.1386 [M + H -H_2_O]^+^.

### 3.2. General Experimental Procedures

Optical rotations were measured on a Rudolph Autopol I polarimeter at the sodium D line (589 nm). UV absorption spectra were recorded with a Varian Cary 100 UV/vis spectrophotometer (Rudolph Research Analytical, Splendora, TX, USA); wavelengths were reported in nm. ECD spectra were recorded with a Jasco-810 spectropolarimeter (JASCO Corporation, Japan). Infrared spectra were recorded in thin polymer films on a Thermo Nicolet Nexus 470 FT-IR spectrophotometer (Thermo Nicolet Corporation, USA); peaks were reported in cm^−1^. NMR data were acquired at 300K on a Bruker Avance DRX-600 NMR spectrometer (Bruke Magnetic Resonance, Germany). Chemical shifts were reported relative to the residual CD_3_OD signals (*δ*_H_ 3.31; *δ*_C_ 49.0) as an internal standard for ^1^H and ^13^C-NMR spectra. The HRESIMS data were acquired on an Agilent 6224 TOF LC-MS (Agilent Technologies Inc., USA), resolution 5000, equipped with an electrospray ionization source. Semi-preparative HPLC was performed on an Agilent 1100 system with UV and refractive index detectors using a YMC Pack ODS-A column (250 × 10 mm, 5 μm) (Agilent Technologies Inc., USA). Commercial silica gel (200–300 mesh) and Sephadex LH-20 (GE Healthcare, Bio-Sciences AB, Sweden). were used for column chromatography (CC). Pre-coated silica gel plates (HSGF-254) were used for thin-layer chromatography (TLC) and spots were detected under UV or by heating after spraying with an anisaldehyde sulfuric acid reagent. 

### 3.3. ECD Calculations

Conformational analysis within an energy window of 3.0 kcal/mol was performed by using the OPLS3 molecular mechanics force field via the MacroModel panel of Maestro 10.2 [22]. The conformers were then further optimized with the software package Gaussian 09 [23] at the B3LYP/6-31G(d) level, and the harmonic vibrational frequencies were calculated to confirm their stability. Then, the 60 lowest electronic transitions for the obtained conformers in a vacuum were calculated using TDDFT methods at the CAM-B3LYP/6-31G(d) level. ECD spectra of the conformers were simulated using a Gaussian function. The overall theoretical ECD spectra were obtained according to the Boltzmann weighting of each conformer.

### 3.4. Antimicrobial Activity Assays

Antimicrobial activity was evaluated against Gram-positive bacteria *S. aureus* ATCC 25923 and Gram-negative bacteria *E. coli* ATCC 25922. MIC values for test compounds were assessed using a 96-well plate format with LB broth (tryptone 10 g/L, yeast extract 5.0 g/L, NaCl 10 g/L, pH 7.2), using a 2-fold dilution method. The test compounds were first dissolved in DMSO at a concentration of 3.2 mg/mL, and this was diluted to 128 μg/mL with LB broth. Then, sequential 2-fold serial dilutions of the mixture were carried out in 100 μL of LB broth, and 100 μL of cell cultures were added to each of the wells. After incubation at 37 °C for 16−18 h, the lowest concentrations that completely inhibited the growth of bacteria were detected by microplate reader for each of the test compounds. All assays were carried out in triplicate.

## 4. Conclusions

In summary, the analysis of genome data of S-4 showed the presence of terpenoid BGCs. Furthermore, two new terpenoids, 1-deoxy-8*α*-hydroxypentalenic acid (**1**) and 1-deoxy-9*β*-hydroxy-11-oxopentalenic acid (**2**), were isolated and characterized from the target strain. The discovery of **1** and **2** confirmed that the pentalenolactone pathway was functional in this organism. Compounds **1** and **2** might be shunt metabolites of the main pentalenolactone biosynthesis pathway. In the bioactivity screening, **1** and **2** exhibited moderate antibacterial activities against both Gram-positive and Gram-negative bacteria. The current results validated the biosynthetic potential of Actinobacteria to produce terpenoids. The application of genome mining in this study facilitated the exploration of Actinobacteria as one of the most promising sources of natural drug discovery.

## Data Availability

The data presented in this study are available in article and Appendix A.

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
