# Peer review of "Isolation and Identification of Pentalenolactone Analogs from Streptomyces sp. NRRL S-4"

_molecules, 2021, doi:10.3390/molecules26237377_

Round 1
Reviewer 1 Report
In the submitted manuscript Li et al. isolated two pentanolactone derivatives 1-deoxy-8α-hydroxypentalenic acid (1) and 13 1-deoxy-9β-hydroxy-11-oxopentalenic acid (2) as a new natural compound. Their structures and absolute configurations were elucidated by analyses of HRESIMS and NMR spectroscopic data as well as time-dependent density functional theory / electronic circular dichroism (TDDFT/ECD) calculations. The compounds have very weak antibacterial effect.
The authors find a biosynthetic gene cluster (BGC) possibly synthesizing pentaleno-11 lactone type terpenes. The main problem of this analysis that the author published this result earlier. The table S1 in the manuscript and Table S3 in the following article “Li, H.; Zhang, M.; Li, H.; Yu, H.; Chen, S.; Wu, W.; Sun, P. Discovery of Venturicidin Congeners and Identification of the Biosynthetic Gene Cluster from Streptomyces sp. NRRL S-4. J. Nat. Prod. 2021, 84, 110-119.” is the same. So it is not a new result. This work just a follow-up work where two new metabolites where identified. Based on this the originality/novelty of the work is very limited.
Regarding the new metabolites it should be noted that pentalenic acid is a known metabolite isolated from many Streptomyces species. Pentalenic acid and compound 1 differ from each other the position of hydroxyl group only. The absolute configuration, NMR assignation etc. of this compound is also known. However, in the manuscript the authors do not even mention this compound. This fact is also reduce the novelty of the work.
For structure elucidation more spectra are necessary. Please provide the IR spectra of compounds. The HRMS spectra with wider m/z windows are also necessary. DEPT or HSQC spectra where the CH2 proton easily distinguishable should also added to the Supporting Information.
Information regarding the purity of the compounds should also provide. HPLC analysis could be necessary.
Due to the above mention facts a very carefully and complett reconsideration is necessary.
Author Response
Response to reviewer 1’s comments:
In the submitted manuscript Li et al. isolated two pentanolactone derivatives 1-deoxy-8α-hydroxypentalenic acid (1) and 13 1-deoxy-9β-hydroxy-11-oxopentalenic acid (2) as a new natural compound. Their structures and absolute configurations were elucidated by analyses of HRESIMS and NMR spectroscopic data as well as time-dependent density functional theory / electronic circular dichroism (TDDFT/ECD) calculations. The compounds have very weak antibacterial effect.
- The authors find a biosynthetic gene cluster (BGC) possibly synthesizing pentaleno-11 lactone type terpenes. The main problem of this analysis that the author published this result earlier. The table S1 in the manuscript and Table S3 in the following article “Li, H.; Zhang, M.; Li, H.; Yu, H.; Chen, S.; Wu, W.; Sun, P. Discovery of Venturicidin Congeners and Identification of the Biosynthetic Gene Cluster from Streptomyces NRRL S-4. J. Nat. Prod. 2021, 84, 110-119.” is the same. So it is not a new result. This work just a follow-up work where two new metabolites where identified. Based on this the originality/novelty of the work is very limited.
The current manuscript is a really follow-up work of the J. Nat. Prod. paper. Table S1 in this manuscript is Table S3 in “J. Nat. Prod. 2021, 84, 110-119.” According to the genome sequencing results, there are 25 BGCs present in the S-4 strain including 6 terpene BGCs. However, only two BGCs metabolites have been identified in S-4 strain. Our recent research interests focus on terpenoids from bacteria. So, we tried to discover new terpenoids on the basis of terpene BGCs information.
- Regarding the new metabolites it should be noted that pentalenic acid is a known metabolite isolated from many Streptomyces species. Pentalenic acid and compound 1 differ from each other the position of hydroxyl group only. The absolute configuration, NMR assignation etc. of this compound is also known. However, in the manuscript the authors do not even mention this compound. This fact is also reduce the novelty of the work.
Pentalenic acid is a well-known pentalenolactone shunt metabolite isolated from many Streptomyces species. Information related to pentalenic acid has added in the manuscript. Similar to pentalenic acid, compounds 1 and 2 are belonging to the pentalenolactone shunt metabolites. Pentalenic acid is generated from 1-deoxypentalenic acid by a cytochrome P450, which is out of the pentalenolactone cluster in S. avermitilis. Because of the difference between S-4 and S. avermitilis, we did not observe the production of pentalenic acid in S-4. Like pentalenic acid, the formation of 1 and 2 might be catalyzed by other P450s present in S-4, which awaits further investigation.
- For structure elucidation more spectra are necessary. Please provide the IR spectra of compounds. The HRMS spectra with wider m/z windows are also necessary. DEPT or HSQC spectra where the CH2 proton easily distinguishable should also added to the Supporting Information.
The IR spectrum of compounds 1 and 2 were added to the Supporting Information.
The HRMS spectra with wider m/z windows were added to the Supporting Information.
The HSQC spectrum and the enlarged region for CH2 were added to the Supporting Information.
- Information regarding the purity of the compounds should also provide. HPLC analysis could be necessary.
The purity of 1 and 2 was analyzed by HPLC as shown in attached file.

Reviewer 2 Report
Comments and Suggestions for Authors
The work of Li et al “The Discovery of Pentalenolactone Analogs from Streptomyces sp. NRRL S-4” reports on the isolation of two new sesquiterpenoids produced by Streptomyces sp. NRRL S-4. The manuscript is well written and concise, but for some instances it is too general.
Overall, the experimental work and interpretations of data are well founded. However, the following concerns and clarifications must be addressed.
- Regarding the fermentation process, the culture medium seems very rich. However, transfering 100 mL culture (in a 250 mL flask), to 10 L bioreactor seems to be not very promising for biomass to proliferate. How (why) the medium is selected? is there any preference for the rich carbon sources, or do their components somehow induce the synthesis of the isolated molecules?
- What was the yield of the new molecules after prepurification? Enough for NMR analysis? The authors are studying seconday (intermediate) metabolites that are part of anabolic processes and that might not be synthesized depending on the induced stress condition. Do the authors consider the pentalonolactone pathway is active to accumulate these sideproducts, or an induced stress is essential? Did the author planned ahead this possibility and based on that did formulate the culture medium?
- How the authors end up having just two samples that are the ones with antibiotic activity and are present in the culture?
- The authors manifest that “The discovery of 1 and 2 confirmed the pentalenolactone pathway was functional in this organism” L 207. However, no pentalenolactone was obtained. If the precursor metabolites are syntesized why the last product is just blocked? If the organism has the pathway for synthesis of pentalenolactone, and part ot it was used to produce the two isolated compounds, what makes the pathway to stop to accumulate side products and avoid the syntesis of the end compound? Is it present but is not enough to be detected? Or it is jus that the pathway might be present but not active under the cultivation conditionsthe cells were exposed to? Why the authors just avoid providing further insights?
- How the deduced function of ORF of the dll BGC compare to the open-reading frames corresponding to the apparent biosynthetic operon for the sesquiterpene antibiotic pentalenolactone?
- Which were the parameters for the antiSMASH analysis? what was the imput data?
Author Response
The work of Li et al “The Discovery of Pentalenolactone Analogs from Streptomyces sp. NRRL S-4” reports on the isolation of two new sesquiterpenoids produced by Streptomyces sp. NRRL S-4. The manuscript is well written and concise, but for some instances it is too general. Overall, the experimental work and interpretations of data are well founded. However, the following concerns and clarifications must be addressed.
- Regarding the fermentation process, the culture medium seems very rich. However, transfering 100 mL culture (in a 250 mL flask), to 10 L bioreactor seems to be not very promising for biomass to proliferate. How (why) the medium is selected? is there any preference for the rich carbon sources, or do their components somehow induce the synthesis of the isolated molecules?
The strain S-4 was inoculated in five 250 mL Erlenmeyer flasks, each containing 100 mL of ISP2 medium, and then transferred with 2.5% inoculation amount to 10 L of fermentation medium. Normally, we conducted a primary chemical screening for the secondary metabolite profiles with several media used in our lab. In the case of S-4 strain, the medium was chosen according to reference (L. Frattaruolo, R. Lacret, A. R. Cappello, A. W. Truman, ACS Chem. Biol. 2017, 12, 2815-2822) and assisted by the results of chemical screening. This medium did contain rich carbon sources. We hope it useful to help the production of secondary metabolites as much as possible. But we are not clear if the components could induce the synthesis of isolated molecules.
- What was the yield of the new molecules after prepurification? Enough for NMR analysis? The authors are studying seconday (intermediate) metabolites that are part of anabolic processes and that might not be synthesized depending on the induced stress condition. Do the authors consider the pentalonolactone pathway is active to accumulate these sideproducts, or an induced stress is essential? Did the author planned ahead this possibility and based on that did formulate the culture medium?
The pentalonolactone-type compounds were obtained with the yields of 0.6 mg/L for 1 and 1.7 mg/L for 2, respectively, which were enough for NMR analysis. The production of pentalonolactone was firstly indicated by the BGC cluster in S-4, showing with a high similarity. Normally, more than twenty BGCs are present in Actinomyces. As to the production of secondary metabolites in each strain, activating conditions are not identical for every BGC. The induced stress with nutrient deficiency is one effective way to active the biosynthetic pathway. Otherwise, rich nutrients in culture medium provide enough precursors i.e., acetyl-CoA, amino acid, which stimulate secondary metabolite synthesis. We did not plan ahead for the pentalonolactone pathway. But we did focus on the terpenoid metabolites which are easy to be overlooked in HPLC analysis due to lack of UV absorptions. As to the selection of culture medium, we usually carried out a primary chemical screening with several culture media in hand.
- How the authors end up having just two samples that are the ones with antibiotic activity and are present in the culture?
Recently, our group are interested in discovering new terpenoids from Actinomycetes. The terpenoids are likely to be overlooked due to low yield and difficulty in detection, which means more opportunity to find new metabolites. The genome sequencing revealed 6 terpene biosynthetic gene clusters in S-4. On the basis of the 10L fermentation broth, we have only found two pentalonolactone family of metabolites in the fermentation broth. The current results could connect metabolites with pentalonolactone biosynthetic gene cluster.
- The authors manifest that “The discovery of 1 and 2 confirmed the pentalenolactone pathway was functional in this organism” L 207. However, no pentalenolactone was obtained. If the precursor metabolites are syntesized why the last product is just blocked? If the organism has the pathway for synthesis of pentalenolactone, and part ot it was used to produce the two isolated compounds, what makes the pathway to stop to accumulate side products and avoid the syntesis of the end compound? Is it present but is not enough to be detected? Or it is jus that the pathway might be present but not active under the cultivation conditionsthe cells were exposed to? Why the authors just avoid providing further insights?
The pentalenolactone biosynthetic BGCs and pathway have been well established by professor Cane’s group. A serious of pentalenolactone analogues and intermediates have been identified in the biosynthesis study. Compounds 1 and 2 are considered as shunt metabolites. The discovery of 1 and 2 suggested the pentalenolactone pathway was at least partially functional. The fact that we did not obtain the last product has puzzled us too. We have compared the pll biosynthetic gene cluster (BGC) and confirmed pentalenolactone BGC. The results showed a high sequence similarity in each of the open reading frames of pll BGC. After consultant with a former postdoctor of Cane. D. E., we realize that the epoxylactone moiety of pentalenolactone is very active and easy to degrade. Cane’ group did very carefully, i.e., at 4℃ or on ice, when dealing with the intermediates purification and in vitro biosynthetic reaction. We did not handle this carefully during the whole fermentation and isolation process. So, the pentalenolactone might be degraded or converted via endogenous biotransformation. The pentalenolactone and lots of its biosynthetic intermediates have already been isolated from many Streptomyces species. Treating with the 10L fermentation broth we only obtained two pentalenolactone family metabolites. Further investigations need re-fermentation of S-4 strain and might result in the isolation of known pentalenolactone analogues.
- How the deduced function of ORF of the dll BGC compare to the open-reading frames corresponding to the apparent biosynthetic operon for the sesquiterpene antibiotic pentalenolactone?
The ORFs of the dll BGC shows > 90% sequence similarity and is identical in organization to those of pnt BGCs. Therefore, we assumed that dll BGC had the complete function for synthesizing pentalenolactone.
- Which were the parameters for the antiSMASH analysis? what was the imput data?
The whole DNA genome of strain S-4 sequenced by the Majorbio company was unloaded to the antiSMASH website: https://antismash.secondarymetabolites.org/. We used default parameters for the antiSMASH analysis. The ORFs of pll and pnt sequence were compared by two sequence alignment on NCBI website to get the sequence identity and similarity.
Reviewer 3 Report
The manuscript under the title of “The Discovery of Pentalenolactone Analogs from Streptomyces sp. NRRL S-4”
- The manuscript (MS) describes the S-4 strain was chemically investigated, resulting in the isolation of two new sesquiterpenoids, 1-deoxy-8α-hydroxypentalenic acid (1) and
- 1-deoxy-9β-hydroxy-11-oxopentalenic acid (2), as shunt metabolites of pentalenolactone (3) biosynthesis.
- This is an interesting study and the authors have collected a unique dataset using several lab tests. Overall the information presented represents valuable information regarding the isolation of bioactive compounds against Gram positive and negative bacteria.
- This version of manuscript is suitable for publication in Molecules Journal with major corrections as fellows.
- Title: The Discovery of Pentalenolactone Analogs from Streptomyces NRRL S-4”, Discovery represent a novel isolated compounds put these compounds were isolated before. So it is better to write: The Discovery of new source of Pentalenolactone Analogs from Streptomyces sp. NRRL S-4. Or Isolation and identification of Pentalenolactone Analogs from Streptomyces sp. NRRL S-4.
- Materials and Methods:
- Fermentation and Isolation must be before General Experimental Procedures.
- Why you select 7 days for isolation of crude extract? Did you have the growth curve for this Streptomyces NRRL S-4? Please explain?
- Why you write 1 and 2 exhibited weak antibacterial activities against both Gram-positive and Gram-negative bacteria. The MICs were look like ok. 16 and 32 μg/mL.
- Check some spelling and grammar errors. Line 33 metabolite profile. metabolite profiles.
- References: Most of references in the MS are after full stop. Which is incorrect.
Author Response
- Title: The Discovery of Pentalenolactone Analogs from Streptomyces NRRL S-4”, Discovery represent a novel isolated compounds put these compounds were isolated before. So it is better to write: The Discovery of new source of Pentalenolactone Analogs from Streptomyces sp. NRRL S-4. Or Isolation and identification of Pentalenolactone Analogs from Streptomyces sp. NRRL S-4.
We have revised the title as suggested.
- Materials and Methods: Fermentation and Isolation must be before General Experimental Procedures.
The section was adjusted as suggested.
- Why you select 7 days for isolation of crude extract? Did you have the growth curve for this Streptomyces NRRL S-4? Please explain?
We did not carry out growth curve for the S-4 strain. Usually, it takes 6-9 days for secondary metabolite production of Streptomyces strains except some special species or purpose. In our lab, we used to collect the fermentation broth at 7-8 days.
- Why you write 1 and 2 exhibited weak antibacterial activities against both Gram-positive and Gram-negative bacteria. The MICs were look like ok. 16 and 32 μg/mL.
As suggested, we revised “weak antibacterial activities” to “moderate antibacterial activities”.
- Check some spelling and grammar errors. Line 33 metabolite profile. metabolite profiles.
We have checked the spelling and grammar errors and made corresponding revisions.
- We have checked the mistakes, spelling and grammar errors in the manuscript and made corresponding revisions .
- We have added discussion related to pentalenic acid and the explain why the final product pentalenolactone was not isolated.
- We have revised Scheme 1.
- We have added IR and MS data to the Supporting Information.
- References: Most of references in the MS are after full stop. Which is incorrect.
We have revised this as required.
Round 2
Reviewer 1 Report
The Authors answered all my questions. The quality of manuscript enhanced. Now it is acceptable for publication.
Author Response
We thankvery much for your kind comments and suggestions.
Reviewer 2 Report
The authors have done a good job. I thank them for clearly addressing my comments and queries.
Author Response
Thank you very much for your kind comments and suggestions.
Reviewer 3 Report
The authors have been responded for all comments and i think the paper now is acceptable
good luck
Author Response

(The authors gave the same response as above.)
